# Overview on Foam Forming Cellulose Materials for Cushioning Packaging Applications

**DOI:** 10.3390/polym14101963

**Published:** 2022-05-11

**Authors:** Petronela Nechita, Silviu Marian Năstac

**Affiliations:** 1Research and Consultancy Center for Agronomy and Environment, Engineering and Agronomy Faculty in Brăila, “Dunărea de Jos” University of Galați, 810017 Braila, Romania; 2Research Center for Mechanics of Machines and Technological Equipments, Engineering and Agronomy Faculty in Brăila, “Dunărea de Jos” University of Galați, 810017 Braila, Romania

**Keywords:** foam forming, cellulose fibers, cushioning, packaging, strength, surfactant

## Abstract

Wet foam can be used as a carrier in the manufacturing of lightweight materials based on natural and man-made fibers and specific additives. Using a foam forming method and cellulose fibers, it is possible to produce the porous materials with large area of end-using such as protective and cushioning packaging, filtering, hydroponic, thermal and sound absorption insulation, or other building materials. In comparison with the water-forming used for conventional paper products, foam-forming method provides many advantages. In particular, since fibers inside the foam are mostly trapped between the foam bubbles, the formed materials have an excellent homogeneity. This allows for using long fibers and a high consistency in head box without significant fiber flocking. As result, important savings in water and energy consumptions for dewatering and drying of the foam formed materials are obtained. In cushioning packaging, foam-formed cellulose materials have their specific advantages comparing to other biodegradable packaging (corrugated board, molded pulp) and can be a sustainable alternative to existing synthetic foams (i.e., expanded polystyrene or polyurethane foams). This review discusses the technical parameters to be controlled during foam forming of cellulose materials to ensure their performances as cushioning and protective packaging. The focus was on the identification of practical solutions to compensate the strength decreasing caused by reduced density and low resistance to water of foam formed cellulose materials.

## 1. Introduction

The protective and cushioning materials are the most important part of package, especially of the fragile goods packaging (i.e., electronics, glass etc.) and as result, they need to be designed and produced extremely carefully. Besides great cushion performance which can effectively protect the packaged products, these packaging materials must have low density to reduce transport cost, barrier properties to avoid moisture, and good process ability to assure an appropriate buffering effect [1]. The first parameter to be considered in choosing of cushioning packaging materials is efficiency of impact energy absorption, and the control of impact acceleration that will be transferred to the built-in product in the range of product fragility. Based on protecting product requirements, these packaging must to perform a greater cushioning efficiency as more shock energy absorption by the unit volume of buffer material as well as a less amount of material for the design [1,2].

The existing and widely used materials for protective and cushioning packaging are mainly produced from plastic foams and expanded or extruded polystyrene. This introduces a major disposal problem for producers and consumers, since these materials are lightweight and bulky and so do not lend themselves to a viable economic and environmentally responsible recycling operation as a result of high handling and transportation costs. In addition, they are not biodegradable, being difficult for soil disposal or composting operations. These are important challenges to accelerate the development of biodegradable and sustainable materials for cushioning applications [3].

Regarding the biodegradable package for cushioning and goods protection, the most currently used are corrugated board, honeycomb board and molded pulp [1]. Compared with EPS (expanded polystyrene) foams and other cellulose-based packaging. Foam formed cellulose materials have their specific properties for cushioning packaging (Table 1).

However, some drawbacks of these biodegradable materials limit their utilization in cushion packaging applications. Although it is widely applied in packaging, used as cushion the corrugated board is not sufficiently soft and easy to abrade the surface of packaged product. In addition, it is water sensitive and presents weak overloaded resilience. Since honeycomb has good buffer performance, their total production cost is high, and some properties don’t meet the requirements, such as poor humidity resistance. The molded pulp materials have good cushion properties, can be used to fix product and its accessories and provide protection. But, until now, its common applications are limited to packaging of goods with low volume and weight such as smart phones, printers and set top boxes. For the heavy products with large size (i.e., washing machines) the buffering effect of these packaging materials is not suitable [10,11,12].

Thus, the development of technologies and technical solutions to obtain novel bio-based and sustainable packaging to reduce petroleum-based ones and to meet the specific cushion packaging requirements is a continuous and high interest activity and emergency issue for the researchers and producers. Being renewable, non-toxic, and with appropriate structural and strength properties, lignocellulosic fibers represent a viable alternative to contribute to these issues [13]. In this context, there are considerable studies and research that have been performed and made a great breakthrough in the field of obtaining the porous cushioning packaging materials based on lignocellulosic fibers (pulp from wood or plant, straw fibers, bagasse fibers etc.) and natural or synthetic polymers (starch, polylactic acid, PLA; polyvynilic alcohol, PVA; polyacrylamide, PAM; etc.) produced by a special method which uses foam as carrier instead of water [14,15,16,17,18]. Based on their eco-friendly features, unlimited market potential, and appropriate properties for packaging applications (i.e., light weight and shock absorbing), nowadays, these materials have become a highly interest for researchers all over the world [2,3,19].

The foam forming process is intensively studied and developed in Finland, at VTT Research Centre [20] where, since 2013, the first pilot plant in the world was implemented to produce the foam formed cellulose materials [21]. During 2021 Stora Enso introduced for consumers testing the material named Papira^®^ obtained from wood pulp foam with lightweight and shock absorbing properties. This is an attractive alternative for protective and cushioning material in packaging [22]. Additionally, these materials are renewable and biodegradable, lightweight and appealing, and they have other important advantages; they can be fully recycled with regular paper and board streams and can have a large variety of end-use applications [23,24] (i.e., protective packaging, thermal packaging, sound and construction insulation, hydroponics etc.). A graphical illustration of these findings is presented in Figure 1.

In this review the focus was on the identification of some factors and technical parameters with influence on the foam forming process of porous cellulose materials, as well as the technological routes to enhance their performances according with the requirements for protective and cushioning packaging.

## 2. Foam Forming of Cellulose Based Materials

The driving force of foam forming process is the increasing the need for finding more cost-efficient alternatives capable to handle a variety of raw materials for the manufacture of sustainable and value-added fibers-based products. The foam is present within large industrial fields as food, petroleum, mining, gas industries, oil recovery, waste water treatment, firefighting, or the textile industry [25]. Aqueous foam can be used as a transfer medium to form lightweight materials with high porosity from cellulose and man-made fibers together with other types of raw materials (i.e., polymeric additives) [1].

Around the 1960s, the foam forming was explored for the first time by Radvan (1964) as based on the replacement of water with foam in paper making process. As result, during this period, the Radfoam process was developed [26,27,28,29]. The principles of this process are attractive, due to their technical and economic advantages: in foam forming a high air content is present (approx. 50–65%) with result on obtaining of fibrous network with bulky structure and outstanding uniformity. Moreover, it is found that this technique enhances the formation properties giving the possibility to use a higher consistency in head box which leads to important reductions of water consumptions and energy costs for dewatering and drying of the foam formed structure [30,31,32,33]. In addition, the aqueous foam is a pseudoplastic material with high viscosity at low shear conditions, but low viscosity at high shear. This feature contributes to improving of fiber mixing and dispersion in the forming section, where, as result of high shear conditions, a reduction of foam viscosity occurs [34,35].

Foam forming technology also provides the possibility of widening the range of product properties, or even generating new ones for paper and board products. This could contribute to the renewal of the paper and board industry. Using this process, it is possible to obtain of porous 3D shapes cellulose-based materials that extend the area of applications, from paper and light weight packaging as was above mentioned, to nonwoven fabrics, insulation, filtering or tissue engineering [24,36,37].

To produce foam, a surfactant is added to aqueous dispersion of fibers, and the mixture is intensively agitated. The bubbles in the foam act as spacer particles between fibers, preventing their flocculation (Figure 2). If the foam is quickly removed via suction, it is obtained a fibers sheet with a greater uniformity than may be achieved in water-based papermaking, in particular when long fibers are used [32,33,34,37,38,39,40]. Comparing with water formation, when a high vacuum it is applied to drainage the suspension and a flat and dense fibrous network is obtained, in case of foam forming the foam is freely or low vacuum drained being possible to obtain lightweight materials with low-density and three-dimensional fibrous network (Figure 3).

Foam forming process enables obtaining of very porous and thick molded structures using new raw materials and combinations, creating the opportunity to open of new markets with sustainable and recyclable bio-based fiber-based products. Low-density property improves both the sound and the thermal insulation characteristics of these materials. In addition, low-density materials can be widely used within applications strictly require lowest supplementary weight added by the protective system. This is limited due to the lack of technical applications and by the poor scientific literature related to this area. In addition, in the last few years, the emergence of scientific studies reveals the innovative character of using bio-derivatives within the insulation domain (such as shock, noise, and thermal) [32].

Another advantage of foam forming is that enables the possibility to use different recyclable or renewable materials as natural fibers, nanoparticles, nanocellulose, flexible long fibers, recycled papers, peat, hemp waste, waste from brewers, etc. [41,42,43,44]. In the Table 2 are summarized the main advantages of foam forming technology compared with water forming process used to obtain conventional paper products.

As is presented in the Table 2, the paper obtained using foam forming process has high bulk and low strength characteristics in un-pressed state compared with water-laid paper. The strength could be enhanced by refining of fibers or by pressing. Pressing raised the strength to the water-laid level in line with bulk reducing [48,49,50].

Although overs last years, researches have been focused on lightweight and low-density fibrous materials also produced by foam forming process. There are still many technical parameters that need to be clarified in this field such as the correlations between the structure and strength properties or water affinity of these fiber foam materials [41,51]. The mechanical strength, especially the compressive modulus it is a very important property for the foam formed materials, especially when are used as protective and cushioning packaging. This type of packaging must to be able to withstand the impacts and shocks or environmental conditions associated with its intended use. In addition, the packaging materials must to have an appropriate strength and structural and barrier properties to protect the packaged products against external conditions (impact to shock, high/low level of temperature or humidity) [24]. On this discussion, the decreasing of strength is the main drawback of foam forming technology.

It has been established that the fibers bonds represent an important parameter with direct influence on the optical and mechanical properties of cellulose fibrous network (i.e., paper). The strength of the fiber bonds depends on the contact area between fibers and has a directly influence on the strength of fibrous network. Therefore, when the mechanical strength properties of fibrous structure need to be improved, the type and area of individual fiber bonds are of particular interest [52]. In case of the foam-formed cellulose materials without strength additives the bonding within the fibrous network is based solely on water induced hydrogen bonds and chemical bonds between the fibers. The strength is a physical property that is influenced by the structural heterogeneity which affects both stress concentration and local load-carrying capacity. The deformation behavior can be sensitive to the size distribution of fibers and fines components within fibrous network. Moreover, for cushioning packaging materials the mechanical property is correlated with force and deformation, which are mainly studied according to the stress-strain curve of materials. According to rheology, cushioning package is a viscoelastic material, characterized by three specific elements such as elasticity, viscosity and plasticity [1].

As mentioned above, the compensation of loss strength caused by reduction of density and water sensitivity is an important challenge in foam forming process and for performances of the foam formed materials. Thereupon, the choice of foaming agent may be critical besides of proper selecting of raw materials. For example, the charges of surfactant molecules can interact with those of fibers and polymers, affecting their bonding. Moreover, the compatibility of surfactant with other process chemicals such as hydrophobization agents has to be ensured. On the other hand, foaming agents such as polyvinyl alcohol (PVA) can also be used as a strength additive and to functionalize the formed structure, which may help to reduce the amount of other chemicals in the process [24].

In conclusion, with an optimal combination and distribution of raw materials and process additives, it is possible to improve the compression strength through local load bearing regions without deteriorating large-scale material homogeneity. It is very important to understand the interaction of foam-fibers, and how the type and morphology of fibers affects the properties of foam formed material. These aspects are discussed in detail in the next sections.

## 3. Technical Parameters with Influence on the Strength and Structural Properties of Foam Forming Fibrous Materials

### 3.1. The Effect of Fines and Micro (Nano) Fibrillar Structures

In the water forming method, the formation of web paper is dependent by fiber length. In this case the presence of fine material influences on dewatering and a good formation can be achieved by including of short fibers in fibrous composition. When a high number of long fibers is used, the formation start to deteriorate and flocculation occurs. In the foam forming the formation is much less dependent of fiber properties, and a significantly better formation was achieved with all pulps than in the case of water- laid paper. The open structure of foam formed sheets enables the high fine material usage. The fine materials reduce the surface tension and improve the bulk and porosity of cellulose foam formed materials [42]. Based on above mentioned aspects, Pohler et al.

The authors of [51] in their research explored the obtaining of a foam formed network structure by combining of wood and hemp fibers with different dimensions and flexibilities. They used fibers based on hemp bast with high content of fines and wood fibers with length between 40 μm–15 mm and width from 3 nm to 50 μm. Supplementary was used two different fibrillated cellulose grades obtained from a never-dried bleached softwood kraft pulp with high content of cellulose microfibrils (CMF), and TEMPO ((2,2,6,6-tetramethylpiperidin-1-yl)oxyl) oxidized cellulose nanofibrils (TCNF). To ensure the retention of fines, they used galactomannan hemicellulose obtained from the seed endosperm of the carob tree and the conventional cationic wet end starch with DS = 0.035. The obtained results emphasized that the resilience of samples after compression was not affected by the bonding properties but only by the dominant fiber type. This suggests that resilience is related to the recovery of fibers from earlier buckling and poor resilience does not necessarily mean that inter-fiber bonds would have failed [51].

In addition, utilization of micro(nano) fibrillated cellulose fibers can improve the strength properties of foam formed fibrous materials, especially when they are used for packaging applications. In this context, Kinnunen et al. [53] have tested in their research six different types of microfibrillar structures (coarser and more refined types), isolated from the wood-based cellulose to increase the strength of foam formed materials made from chemical and mechanical pulp. The results show that by adding of micro-fibrillated cellulose (MFC) to the fibrous suspension, the mechanical strength of materials is improved without affecting their bulk. However, in case of Scott bond, the coarser MFC gave slightly higher value compared to more refined MFC. On the other hand, more refined MFC gave higher tensile index value. This example illustrates that when there are good water drainage properties it is possible to use high addition levels of MFC as strengthening additive that can leads to obtaining of high bulky products with an adequate strength value [42].

In their work, Sormunen et al. [52] highlighted that the strength of inter-fibers bond and contact area between fibers can be improved using fibrillar and fine material. In their opinion, the improving of strength acts by two mechanisms. The first one is based on the fact that the fine material contributes to increase the number of bonds and reduces the mean fiber segment length building the loose materials inter-fibers bridges as result of surface tension forces during drying. In this case a higher density material is obtained. In the second one, the strength of individual bonds is affected by the presence of fibrillar components which enhances the mechanical entanglement of the inter-fiber fibrils or changes these chemical forces. By this way the larger contact area is created [54].

In their study Ketoja et al. [55] proposed a new buckling theory including a statistical distribution of free-span lengths. The tested lightweight materials with the density range between 20 kg/m^3^ and 100 kg/m^3^ were prepared by use of: (1) wood fibers from chemo-thermomechanical pulp (CTMP) and bleached softwood Kraft pulp (BSKP) and (2) viscose fibers reinforced with the cellulose nanofibers (CNF) additive. In their experimental program they varied the sample density, method of fibers bonding (via wood fibers or nanocellulose), fiber stiffness, bond properties (degree of fibers refining), and surfactant type. The theory predicts universal ratios between stresses at different compression levels for low-density random fiber networks. The buckling of fiber segments dominates the strength behavior and this model can be used in developing the properties of lightweight materials in novel applications.

Y. Liu et al. [56] have studied the structure of foam formed cellulose materials based on bleached hardwood Kraft pulp, softwood acid sulphite pulp and bagasse pulp reinforced with microfibrillate cellulose (MFC, particle size of 60 μm; bulk density 0.3 g/cm^3^). Sodium dodecyl sulphate (SDS) (≥98.5%), polysorbate 80 (Tween-80) and beeswax (refined) were used as surfactants. A cationic polyacrylamide (CPAM) with high molecular weight and low charge density was used as wet strength additive. The obtained results showed that the SDS/MFC system, Tween-80 system, and beeswax system generated the foams with porous microstructure. Regarding the compression strength of pulp foam structure, the results showed that the SDS pulp foam was extremely weak without MFC addition. When MFC was used as reinforcing, SDS/MFC system exhibited superior mechanical properties comparing with other surfactant systems. Cervin et al. [57] obtained strong porous nanofibrillated cellulose (NFC) foam, prepared from surface modified cellulose nanofibrils. They assessed the influence of low NFC concentrations on the foam stability in an aqueous suspension. Comparing with non-stabilized foam, the lifetime of the aqueous foam is significantly prolonged. Upon drying the foam yields a porous material with an average pore diameter of 500 μm which has low density (30 mg/cm^3^) and 98% porosity. These porous materials have superior values of Young’s modulus in compression (737 KPa) compared with other cellulose foams obtained by freeze-drying but are lower than those of polystyrene foams. The compressive energy absorption value has 48 kJ/m^3^ at 80% strain, similar with those of cellulose aerogels and freeze-dried cellulose foams.

Li et al. [58] prepared high performance porous foams based on nanofibrillated cellulose (NFC) and ethanol, isopropanol, tert-butanol, and n-butanol followed by freeze-drying. They found that the foams prepared from NFC suspensions containing ethanol, isopropanol and n-butanol exhibited highly porous structures with a honeycomb-like cellular texture and well-defined pores comparing with foams based on tert-butanol/NFC, which exhibit a higher number of smaller size pores with irregular shape. The foams prepared by freezing at −196 °C with ethanol also revealed small size pores, with no layered pore structure. The results obtained suggested that freeze-drying could be used to control the key foam parameters by adding different alcohols into NFC suspension and adjusting the freezing temperature. In their research, Liu et al. [59] used cellulose nanofibrils as reinforcement for freeze—drying biodegradable foams based on polyvinyl alcohol (PVA) and established the correlations between their microstructure and properties. The results show that the nanofibrils content, the freezing temperature and the solid contents of the precursor suspension have influence on the microchanneled structure of foams. In this context, the size of foam porous channels increases at high content of cellulose nanofibrils but it is decreased as the freezing temperature is lower and the solid content of suspension is higher. At 30% cellulose nanofibrils content it is obtained a material with high porosity and improved compressive strength. The using of cellulose nanofibrils in the foam composition not only reduced the water absorption and improved the dimensional stability of PVA foam but also accelerated the biodegradation rate of foam materials with promising recommendations for application in packaging field or as energy absorbents.

The above presented results suggest that the fine material improve the foam formation and introducing of nano(micro) fibrillated cellulose in the composition of foam formed cellulose materials demonstrated the obtaining of material with strong mechanical strength in spite of its light weight and high porosity. This enlarges the potential of using fibrous cellulose foams as structural insulation materials, templates for inorganic nanoparticle synthesis, energy absorption applications, as packing materials, and even applications based on the material porous structure such as filtration, adsorption, drug delivery, and catalysis [60,61,62,63].

### 3.2. The Polymer Additives Charge and Type

It is known that the utilization of polymeric additives is a technical route to improve the wet and dry strength of cellulose-based materials. There are many researches where the effect of strength polymeric additives is analyzed for the foam forming cellulose materials, also. Thus, Meiyan et al. [64] obtained the foam formed materials from bleached softwood Kraft pulp (BSKP) and sodium dodecyl sulphate (SDS, purity 86.0%) as foaming additive. To improve the mechanical strength, cationic polymers such as chitosan polysaccharide (10–20%) and polyacrylamide-CPAM (0.3–2.0%) were used as reinforcing of borate-crosslinking cellulose foams. The obtained results show that a high mechanical strength and good thermal insulation (thermal conductivity 0.068 W/(m.K)) have been obtained for the low density samples (52.65 mg/cm^3^) with 20% chitosan and 0.5% CPAM content. The properties were comparable with the commonly and commercially used fireproof mineral wool, cellular inorganic materials and foam glass products. In addition, the obtained cellulose foam formed materials exhibit a better fire resistance, antibacterial properties, and sound absorption as result of the synergistic actions of borate, chitosan and CPAM. The boron ions could crosslink with hydroxyl groups of cellulose and chitosan to form strong covalent and hydrogen bonds. Furthermore, the electrostatic interactions between cellulose fibers and chitosan or CPAM lead to the excellent mechanical properties.

According to the results obtained by He et al. [65] the compression strength and the recovery after compression of the bleached softwood pulp foams were significantly improved using an adapted borate cross-linking approach. In this case, by introduction of 1.5 wt% borate, the compression strength of the cellulose foam could reach to 74.1 kPa, which is 28 times higher than that of reference sample. The cross-linked cellulose foam could be compressed to large strains and subsequently expand back to its original shape. The height recovery of the cellulose foams was evaluated by unloading the compressed specimens and after six cycles of compressing the releasing 90% of their initial heights were maintained. The authors considered that this behavior is attributed to the covalent bonds between borate and cellulose fibers which increase the strength of cellulose foam. In opposite, the excessive borate content (>1.5 wt%) deposited on the surface of pulp fibers could reduce the interaction force between fibers decreasing the strength of cellulose foam materials. For example, when the borate content is increased from 1.5 to 2%, the compression strength of pulp foam is reduced from 74.1 to 37.3 kPa. Regarding the behavior of foam formed cellulose materials at compression and the recovery properties, Järvinen et al. [66] suggest that these materials exhibit unique recovery properties related to the fiber network structure rather than the lower water surface tension or improved dewatering. 

In their research, Seppanen et al. [67] and Vähä-Nissi et al. [68] obtained the improving of wet tensile index and resistance against disintegration in water of foam formed cellulose materials by utilization of polyamidoamide-epichlorohydrin (PAE) as additive. The promising results regarding the improving the wet strength, were obtained with bio-based and/or biodegradable alternatives, especially chitosan + poly(ethylene maleic acid) PEMA, also. Ahmadzadeh et al. [69] introduced surface modified montmorillonite (SM-MMT) in composition of cellulose nanocomposite foams for food packaging application in order to reduce the bubble sizes and to improve the mechanical and barrier properties as well as the thermo insulating performance. The results show that, due to the reduction of average cell size, the presence of montmorillonite improves the thermal insulation properties. In addition, comparing with pure cellulose foams, both mechanical and barrier properties of nanocomposites were significantly enhanced at low content of montmorillonite. Based on above mentioned results, these materials can be used as a possible alternative to expanded polystyrene (EPS) foam trays currently used for dry food packaging.

In their work, Liu et al. [70] have studied the influence of polyvinylalcohol (PVA) and sodium tetraborate content on the mechanical and static cushioning properties of bagasse cellulose foam formed materials. The obtained results emphasized a good compatibility of PVA with cellulose fibers. By adding of PVA the mechanical properties are improved and the number and diameter of pores within fibrous structure as well as their thickness wall are increased. When the PVA content is 10.91 wt%, the comprehensive mechanical properties and static cushioning properties are closest to those of high density expanded polystyrene (HDEPS). The similar results were obtained by the increasing of tetraborate content but in this case the added amount cannot exceed 0.109 wt% due to the increasing of the degree of gelation with negative effect on the cell stability. The cushioning properties of foam formed materials are influenced by the water content. The water acts as plasticizer by adjusting the hardness and cushioning performance. If the amount of water added is too small, the material is too hard, and if the amount of water added is too large, the cell strength of the material is lower and the deformation is large. Additionally, these new foam-formed materials are cheaper and biodegradable, fulfilling the requirements demanded by environmental issues. Nevertheless, both matrix materials contain hydroxyl groups. Under high humidity conditions, hydroxyl groups absorb water, resulting in a reduction of cushioning performance. The authors agree that a suitable waterproof coating is needed to protect the cushioning material.

Luo et al. [19] explored the effects of the initial water content on the foaming quality and mechanical properties of wood poplar-fiber-based material for cushioning packaging. As foaming agent was used azodicarbonamide and other additives were sodium bicarbonate, starch, and nucleating agent-French chalk. The obtained results showed that the initial water content had a substantial influence on the foaming quality and the mechanical properties of foam formed porous materials. They obtained the optimal bubbles growth, porosity, and size and distribution of pores when the initial water content was about 69.3%. In these conditions the best mechanical properties of the samples were registered. Li et al. [3] studied the impact of mass ratio of fiber to starch, the plasticizer content and foaming additives on the compressive strength of biomass cushion packaging material based on straw fibers. They obtained an optimum comprehensive strength (about 0.94 MPa) and the best cushioning performance at 2:5 mass ratio fiber: starch, 12% plasticizer content and 0.1% foaming additive.

Ottenhall et al. [71] combined cellulose fibers with cationic polymers (cationic polyvinylamine and deacetylated chitosan) to obtain the lightweight and low-density foam materials with good water-stability and good antimicrobial properties. To improve the water stability of foams the citric acid was used. To the authors’ knowledge, this is the first instance when the citric acid and chitosan has been used to produce a water-stable low-density cellulose fiber foam material with antimicrobial properties. Comparing with freeze-drying or organic solvents processes [72,73,74] this method is simple and facilitates large scale production. The obtained results show that the produced foam material containing only bio-based polymers (i.e., cellulose fibers and chitosan) had water-stability and appropriate antimicrobial properties. This can be a good alternative for the packaging of fragile products that need to withstand both moisture and microbial attack. All of the foams were resistant to fungal growth under humid conditions; but only the foams containing chitosan inhibited fungal growth when nutrients were present. The cationic chitosan provides both antibacterial and antifungal properties, and it can be used as a contact-active antimicrobial component. This could be a future alternative for the environmentally sustainable low-density packaging materials that can withstand microbial growth while protecting sensitive products against mechanical damage [75,76].

In their experiments, Paunen et al. [77] observed that the recovery after compression of lightweight cellulose materials with 3D network can be improved by addition of elastic polymers that accumulate at the fiber joints during foam forming and drying operations. They considered that although the recovery after deformation is elastic, a large proportion of the return towards the original dimensions is influenced by time-dependent creep recovery of the material. In addition, the utilization of an elastomer with high stiffness and uniform distribution over the material improve compression recovery.

As mentioned above, the utilization of polymeric additives improves the mechanical and cushioning performances of foam formed cellulose materials. However, to meet the environmental requirements (i.e., biodegradability) the biopolymers (e.g., starch, chitosan) can be an appropriate alternative to petroleum-based polymers as polyacrylamide or polyamidoamide-epichlorohydrin.

### 3.3. The Morphology and Refining of Cellulose Fibers

Generally, the most of additives are synthetic products which create environmental issues and the procedure of their utilization is costly. Additionally, these additives are absorbed on fibers through hydrogen bonds and are easily removed during dewatering and drying. As result, the mechanical properties of porous materials are reduced [78]. As above mentioned by reinforcing of cellulose foam formed materials with nanofibrillated or microfibrillated cellulose combined with freeze-drying, materials with small pores, uniform structure and high strength can be obtained. However, the cost of these reinforcing additives is very high (about 1750–20,000 $/Kg) and the freeze-drying curing process is energy intensive, that limits the applicability of this technology for a large-scale production [79]. Therefore, the inexpensive and easy accessible treatments to enhance the mechanical strength of cellulose foam formed materials must to be identified.

A specific feature of cellulose fibers is their filamentous structure which contributes to development of hydrogen bonds. This structure has potential to develop of reticulated open cell network which is specific for foam materials. It is known that the mechanical strength of conventionally paper materials it is mainly affected by the fiber length and the extending of bonding area between cellulose fibers, which can be improved by the mechanical treatment of fibers using beating or refining processes. In addition, the absorption capacity of fibers is directly proportional with their surface area, which can be improved by fibers refining. Therefore, it is important to modify the fibers surface to obtain a better contact bonding among them. For example, for a fibrous network with the density of 300–1000 kg/m^3^ the number of inter-fibers bonds per each fiber is large comparing with low density (20–100 kg/m^3^) materials which have a low relative bonded area and the number of inter-fibers bonds is greatly reduced. The fibers heterogeneity becomes essential to carry the compressive stress in case of fiber network with a low relative bonded area. Generally, the fibrous network is based on the fibers with a small diameter and a high aspect ratio as main component. As result, the internal structure of the fiber assembly, their properties, contacts among them and the dimensions of fibrous network influence the response of material under loading [55]. In this context, Chen et al. [78] tested this theory for preparation of ultra-low density pinus fiber composites obtained by foaming process having in composition aluminum sulphate and sodium silicate as inorganic fillers. The optimum values for mechanical strength (i.e., tensile index: 109.4 kN/m, burst index: 577.0 kPa, and tear index 58.0 mN) was obtained at a beating gap of 30µ. The presence of aluminum sulphate and sodium silicate contributes to the breaking of the intramolecular hydrogen bonds and forms covalent bonds as Si-O-C and Al-O-C within the structure of ultra-low density composites. At 35°SR beating degree (beating gap of 30 µ), the internal bond strength of foam formed structures was of 50.9 kPa, with about 73% larger than composite structures obtained at a beating degree of 13°SR. This is the result of advanced fibrillation of cellulose fibers by increasing of beating degree. Therefore, the degree of fibrillation at 35°SR was 0.742% comparing to 0.362% at 13°SR.

In their research Smith et al. [79] have studied the tensile properties for both water and foam-formed fibrous materials (sheets). They obtained the water formed sheets with 217 kg/m^3^ density and foam formed sheets with 185 kg/m^3^, respectively. The results regarding the strength properties shown that the foam formed materials had approximately a half of the strength of the water-formed sheets. This decreasing is attributed to the reduction of consolidating forces during drying as result of existing surfactant in the structure of foam formed materials. Though the bulk of foam formed structure is reduced by wet pressing and beating of fibers, the authors considered that these are the appropriate and effective routes to regain the strength of foam formed fibrous materials.

In their study Li et al. [80] assumed that common cellulose pulp fibers can be used as a widely available and inexpensive raw material, without any additives/matrices to obtain foam formed materials with appropriate mechanical strength. They postulated that by beating of cellulose fibers it is produced a controlled number of smaller fibrils (fines) which can connect with the backbone fibers via hydrogen bonds to obtain a fibrous network with improved structure of pores and mechanical strength. By refining of bleached Kraft eucalyptus pulp at 60°SR, they obtained a foam formed material with robust network microstructure which was still mainly composed of large fibers as the backbone, while the fibrillation treatment enabled additional hydrogen bonds between the fibers and fine material. As result, a low density and high porosity is obtained, the pore size distribution became uniform and the compression and cushioning properties are improved. However, when the beating degree continued to increase over 60°SR, the opposite undesired effects occurred, apparently due to the fibers shortening or excessive fiber interactions (i.e., flocculation) [80]. When the cellulose fibers are unrefined a weak or absent of inter-fibers bonding occurs. As result a bulky and weak foam formed structure is obtained (Figure 4a). By contrast, after the fibers refining the surface of bonding area is increased as result of external fibrillation and a more uniform foam material is obtained with improved structural integrity (Figure 4b). Additionally, after drying, the small bubbles of foam are translated into the pores of similar size, separating the fibers in the ultimate foam material.

Madani et al. [81] obtained low density cellulose-based foam, as low as 10 mg/cm^3^ and improved the strength properties according to industrial applications, without mechanical pressing. Based on the obtained results, they considered that the fibers refining is a very important parameter that influences both the number of inter-fibers bonds and formation of foam fibrous network. As the ratio of the refined pulp is increased, the bulk is reduced and the strength is improved. In the standard paper, the improving of strength is based on the individual fiber and network strength. It was shown that as with standard paper, the strength of the foam is independent by the strength of individual fibers as the network fails before intrinsic failure of the fibers. In this case, during mechanical loading failure occurs within the network, and due to the extreme bulk, the individual fibers remain intact. Thus, longer fibers contribute to achieving low density while fine material or highly fibrillated fibers contribute to the strength improving in the expense of bulk. The results obtained by Burke [32] in his thesis, shown that the density and compression strength of foam formed cellulose materials can be controlled via fibers properties and liquid fractions of the dispersions (consistency) during production as well as the duration of drainage. The fiber length influenced compression along z-direction. Thus, longer fibers buckle first followed by the shorter ones which require additional levels of stress to buckle.

Jahangiri, et al. [82,83] analyzed the effect of fiber morphology and crowding number on the filtration and acoustical parameters for the air-dried foam cellulose materials obtained from different ratio of hardwood and softwood pulps refined in Valley hollander comparing with freeze dried foam cellulose materials from nanofibrillated Lyocell fibers. The obtained results shown that increasing of specific surface of fiber and crowding number correlated with decreasing of pulp freeness improve the filtration efficiency and pressure-drop of foam formed pulp samples. The mechanical properties and filtration performance are improved for foam cellulose samples with 10% to 30% weight ratio of softwood beaten fibers in the structure. Moreover, the freeze-dried foam samples from nanofibrillated fibers at high foam air contents show excellent submicron filtration properties and very poor strength properties. The acoustical properties of freeze-dried foam-formed cellulose materials using nanofibrillated fibers are higher than those obtained for softwood foam cellulose materials.

Alimadadi et al. [40] obtained the networks of 3D fiber orientation using refined thermomechanical pulp fibers and foam forming method. Comparing with 2D fiber network the foam formed materials have unique properties being extremely bulky (190 cm^3^/g) and low density (5 kg/m^3^) but with appropriate structural integrity. They found that a 3D oriented fiber network requires much less fiber-fiber contact to create a connected network than a 2D oriented network. These properties are the result of fiber orientation in z-direction (out-of-plane orientation) within the fiber network. To obtain the 3D oriented fiber network, in the foam must to be created a 3D fiber orientation and then maintaining of this orientation during forming, pressing and drying processes. Comparing to other fibrous materials the nonlinear compression stress-strain relationships is also unique. The 3D oriented fiber network exhibits a much higher deformation recovery comparing with paper or other nonwoven sheets. This behavior is due to the fact that in the fiber network there is a wider 3D fiber orientation distribution and a high proportion of very large open pores [77].

In their review, Hjelt et al. [24] emphasized that for the materials with high porosity and low density, the compression behavior is often the most important strength property. The compression modulus and stress are strongly influenced by the density of material besides of bending stiffness of the fibers and their bonding. When the stiffer fibers are used (i.e., chemico-thermomechanical pulp—CTMP) a much higher compression stress is obtained compared with flexible fibers (i.e., Kraft cellulose fibers). For both types of fibers the stress increases with density. This is explained by the increasing number of inter-fibers contacts.

Koponen et al. [13] analyzed the foam forming of cellulose materials using long fibers based on bleached unrefined hardwood kraft pulp (BHK) with 18°SR, bleached unrefined softwood kraft pulp (BSK), with 13°SR, and man-made Lyocell fibers of 6 mm length (TENCEL^®^). The average length in the fibrous mixture was between 1.0 and 3.0 mm. The obtained results show that foam formed BHK and BSK samples have an excellent formation. It is also observed that foam forming gives very good formation even when the mass fraction of 6 mm long Lyocell fibers is 20%. The improvement of formation with decreasing foam density becomes more significant when the average fiber length is increased (Figure 5). It was found that the fiber length had opposite effect on the tensile index and the z-strength. As expected, the tensile index increases as fiber length is increased. However, z-strength decreased with increasing fiber length. This might be due to the lower bonding tendency of Lyocell fibers compared to BHK and BSK fibers. To enable a significant increase of the fiber length, in foam forming process it is possible to use a mixture of wood fibers and natural or man-made long fibers and to increase the solid content after press section (it is was 8% higher comparing to water forming). As a result, savings in drying energy as much as 30% were obtained.

As was highlighted in the Section 2, the foam forming technique allows for the utilization of waste of fibrous materials with a wide range of particles size and fiber lengths, which can be distributed within foam-fiber dispersion. In the performed experiments, Burke [32] used waste from beer production (beer spent grains) in a mixture with kraft fibers to obtain beer coasters by foam forming method. To increase the stiffness and bond of the fibrous structure, polyvinyl acetate was used as additive. The level of mechanical properties was comparable with those of water formed. Nechita et al. [47] obtained the foam formed cellulose materials using bleached hardwood cellulose fibers and recycled fibers from recovered papers, with sound absorption performances comparable with expanded/extruded polystyrene materials. Based on the obtained results they concluded that the high quantity of fine material and short fibers within internal structure of foam formed materials with a high proportion of recycled fibers with small diameters, creates a large tortuosity medium which contributes to a better sound propagation and high sound absorption coefficient comparing with foam formed materials from virgin cellulose fibers (Figure 6). Debeleac et al. [48] performed the computational investigations on these types of foam formed cellulose materials and obtained results provide extended soundproof characteristics to the incidence angle of the acoustic field. In addition, the results supply supplementary information useful for future analyses regarding the influences of random geometry air inclusions into the foam formed fibrous network. 

Năstac et al. [2] evaluated the shock absorption properties of foam formed cellulose materials with recycled fibers in composition. Based on values of restoring coefficient which was in range of 0.8 … 0.9, meaning approx. 90% of the average value from polystyrene and polyurethane foams and impact strength. The authors concluded that these foam-formed cellulose materials demonstrate a good capability related to the shock absorption (Figure 7). Besides this behavior, Park et al. [49] found that the recycled fibers based on corrugated board give a high foamability in the fibrous suspensions comparing with virgin cellulose fibers and formed foam had more bubbles with smaller size.

### 3.4. The Influence of Surfactant Type and Concentration

The compensation of strength decreasing by reducing density of the fibrous structure remains the main challenge in the foam forming process. The choice of foaming agent may be one of the critical parameters with influence on the foaming process and foamed product properties. For example, the inter-fibers bonds are affected by the interactions between electrical charge of surfactant molecules and those of fibers and polymers. In addition, the compatibility of surfactant with other chemicals used in foam forming process (i.e., hydrophobization agents) has to be ensured. Generally, in foam forming process the utilization of suitable surfactant or foaming additive is based on some properties that will be developed during foam forming: to allow a rapid production of foam and to fulfil the requirements in terms of air content (aprox. 65% *v/v* air) and bubble size distribution; to be effective regarding costs; to be available and easy to use; and to meet basic requirements regarding the environmental issues (i.e., biodegradability and toxicity) [84,85]. 

Mira et al. [41] investigated the effect of surfactant concentration for a fiber-surfactant system and analyzed the chemical interactions between foaming additives and cellulose fibers in foam forming process. They tested seven types of surfactants (anionic and nonionic) in chemo-thermo-mechanical (CTMP) and kraft pulp fibrous systems containing ground calcium carbonate (GCC) and cationic polyacrylamide. Foaming experiments of selected surfactants suggest that the rapid foaming is related to the presence of mixtures of surfactants with the right molecular structures and in the right proportion with respect to each other (in the case of commercial SDS, it is the small amounts of dodecanol providing the boosting effect). It was found, also, that the minimum surfactant concentrations required to reach the target foam volume (regardless of whether the foaming was rapid or slow) were lowest for surfactants with anionic character. In addition, the type of pulp fiber and the presence of GCC in the surfactant/pulp system were found to have very little effect on the foaming performance of the suspensions. Studying the effect of type and concentration of surfactant on the quality and mechanical strength of foam formed cellulose materials, Lappalainen et al. [86] found that, in the presence of ionic polymers, the charge of surfactant has a significant effect on the formation of the material sample. Comparing with the water formed cellulose materials the absorption properties of foam formed materials were lower. Furthermore, a high dryness of samples is obtained by foam forming comparing with water forming method. In the presence of ionic polymers, the type and amount of surfactant highly influenced the dewatering process and formation of the material sample. Foaming agent amount had no effect on the mechanical properties of the samples. Other strength additives increase the strength of foam formed materials in the presence of non-ionic surfactants [87,88].

Ran et al. [89] studied the mechanism of foam forming process and the effect of sodium dodecyl sulphate (SDS) as well as the formed bubbles on the properties of foam formed cellulose materials based on cotton linter pulp. They prepared ultra-lightweight cellulose foams by adding SDS to NaOH/urea aqueous solution using intensive mixing. After SDS adding, a foam structure with micro and nanopores is formed and the bubbles about 20–100 μ were observed in the solutions. As result, the cellulose foams exhibit ultra-low density (about 30 mg/cm^3^) and high specific surface area, which are suitable in catalysis, sensing, separation and filtration. Bubbles inside the wet foams lead to lower mechanical properties but inside the dry foams have an opposite role, revealing a good compressive strength. Lehmonen et al. [90] have analyzed the influence of surfactant amount, vacuum level and wet pressing on the forming process and properties of fibrous foam materials from bleached chemical softwood pulp comparing with those obtained in water forming. The presence of foam contributes to a superior dryness of foam formed materials after forming phase. This can be explained by the one hand that at the end of forming there is a less water to remove from fibrous network and air flows fast. On the other hand, the air bubbles from foam remove the water from the sheet better than pure air flow. This behavior can be related to improve the compression dewatering and displacement dewatering. In addition, displacement dewatering may also be improved in foam forming due to bubble surfaces wiping the droplets away from fiber surfaces. The result is an increasing of large pores number in the fibrous foam formed network.

The high surfactant dosage improves the structural properties of foam formed cellulose materials. At a high content of surfactant, the foam bubbles are strong and stable and effectively reduce the fibers’ agglomeration, resulting in improved formation. The density of the foam-formed materials decreased linearly with increasing of surfactant dosage at the wet pressing pressures of 0–50 kPa. Density is strongly related to the structure and strength of fibrous network as it correlates well with the number of contacts between fibers. The foam bubbles limit the possible locations of the fibers, resulting in more open pores but also in more fiber contacts at the areas in-between the bubbles (Figure 8).

Regarding the strength properties of cellulose foam formed materials was observed that the z-strength of dry materials decreased with increasing of surfactant dosage and the foaming process. By decreasing the density and the number of interfiber contacts, a reduction of the individual contacts strength is obtained. The effect is reduced by the presence of foam bubbles and the wet pressing due to the pressure overriding the surface tension forces. The tensile strength and strain at break properties are affected by both the surfactant dosage and the presence of foam similarly to the z-strength [90].

The foam forming of cellulose-based materials is influenced by the surfactant type, also. In this context, Gottberg et al. [91] investigated the utilization of polyvinyl alcohol (PVA) as foaming additive on the foam forming paper. The obtained results show that the degree of hydrolysis of PVA had a stronger effect on foaming comparing with its molecular mass. The foamability is strongly decreased when the degree of hydrolysis increased from 88 to 98 mol−%. However, it is observed a slowly increase of foam stability and bubble size with increasing of molecular mass without influence on maximum air content. Additionally, when a fully hydrolyzed PVA is used, the best strength properties of foam formed cellulose materials were achieved. Comparing with anionic foaming additive (i.e., sodium dodecyl sulphate) the strength properties (both in- and out-of-plane) of foam formed samples were better when was used PVA. It is known, that PVA has binder ability, and when is added to cellulose fibers, is able to further enhance the strength properties of paper and board [92].

Generally, in the aqueous solution, the anionic surfactants (i.e., SDS) tend to precipitate as salts, with calcium and magnesium ions from water. In these conditions, the residues of precipitate increase during foam forming process and affect the quality of the foam-formed materials. In their research, Viitalla et al. [93,94] have tested the mixture of anionic surfactant (SDS) with non-ionic or a zwitterionic surfactant (Tween 20) in proportion of 77/23 (mol/mol), to obtain the foam formed cellulose hand sheets using the hard water with high content of calcogenic ions. The results show that, besides of inhibitory effect of precipitate residues, this mixture of surfactants gave higher tensile index compared to the paper sheets produced using only SDS or Tween 20. Ketola et al. [95] have studied the surface interactions between fibers and bubbles foam in simplified fiber-foam systems with the captive bubbles and fiber bed methods using sodium dodecyl sulphate (SDS) solution as surfactant. The results emphasized that the foam formed structures based on natural fibers is sensitive to both type of fiber and type and concentration of surfactant. The addition of SDS decreases attraction bubbles-fiber surfaces. At high SDS concentration, the attraction was lost due to wetting transition. However, a smooth cellulose fiber surface did not attract bubbles even in the absence of surfactant. This is caused by the presence of hydrophobic regions on the cellulose fibers surface (e.g., lignin) [96,97]. The type of foaming agent may affect the tensile strength by changing the strength of inter-fiber bonds or the network structure [98]. For laboratory Kraft sheets with density of 200 kg/m^3^, Al-Qararah [42] found a slight reduction in tensile strength at high SDS surfactant concentration comparing with water-formed structure in similar conditions.

### 3.5. The Water Affinity of Cellulose Foam Formed Materials vs. Their Cushioning Performance

It is known that the cellulose fibers exhibit sensitivity to humidity and water. Therefore, the improving water resistance of foam formed cellulose materials to prevent the deterioration of package itself and packaged products quality, is another challenge in foam forming process.

Due to the large number of hydroxyl groups in their chemical structure, cellulose fibers exhibit unique hydrophilic character but a high availability for chemical modification to improve their hydrophobicity to be suitable for different applications (i.e., packaging, printing etc.). The traditional hydrophobization solutions consist on using of coupling agents such as anhydrides (i.e., alkenyl succinic anhydride, ASA; alkyl ketene dimer, AKD) and isocyanates as the reactive function, but also others such as carboxylic acids, acyl chlorides and many more. However, with these treatments the fiber surface available for the reaction is limited both in organic solvents and in the dry state. When dried from water, fiber wall is collapsed (due to strong attractive capillary force caused by high water surface tension) and internal structure not accessible. Even if hydrophobized, fibers swell in water. This non-reversible fiber collapse is typically referred to as hornification. To prevent this phenomenon and obtain the non-collapse dried cellulose fibers, the water content of never-dried pulp has to be substituted with a low surface tension fluid (i.e., ethanol) by means of solvent exchange [99,100,101]. In the case of cellulose foam materials, rendering of hydrophobic fibers has been greatly sought and many hydrophobization techniques have been investigated.

In this context, Tejado et al., [5] and Chen et al. [102] prepared the cellulose foam with hydrophobic properties using the chemical vapor deposition of trichloromethylsilane. The method is based on never-dried cellulose pulp which was suspended in water and, through multiple solvent cycles, was suspended in anhydrous ethanol. This allows drying as low-density foam. By reaction with trichloromethylsilane this material rendered hydrophobic. The time and temperature of hydrophobization reaction have been set at 5 min and 60 °C. The contact angle of the obtained material was 150° and the paper sheets prepared from these cellulose fibers exhibit a negative dry-shrinkage coefficient. A large increase of sheet thickness upon drying is obtained as result of expanding to over 500% the original thickness in the first cycle of use. In their research, Zheng et al. [6], obtained cellulose foam formed materials made from bleached chemico thermo mechanical pulp (CTMP). To enhance water resistance, the foams were impregnated with hydrophobic extractives from the outer bark of birch. After drying the water resistance of cellulose foams was improved, and the materials absorbed 50% less moisture within 24 h compared to the unmodified structures. In addition, after immersion in water for 7 days the modified cellulose foams did not disintegrate, showing the increased water resistance. Su et al. [4] described different methods to improve the barrier properties (water and water vapors, oil and grease, oxygen) of cellulose-based packaging including cushioning grades. They found that the coating of cellulose foam materials with additives based on latexes, bio-polymers, cellulose regenerated films, esters of nano(micro)fibrillated cellulose and hemicelluloses can be a sustainable alternative to improve the water resistance of these materials and to replace the existing plastic packages.

It can be concluded that the utilization of one of the above-mentioned methods, introducing the strength additives in fiber suspension or surface coatings of foam formed cellulose materials with hydrophobic compounds, makes it necessary to obtain additional strength and suitable properties according to the end use requirements.

## 4. Conclusions

In this paper, an overview on the process of foam forming cellulose materials is presented with the focus on some of the technical parameters which affect the structure and their functional properties.

The foam forming is an innovative technology that opens new challenges for the pulp and paper industry in the field of porous materials with three dimensional structures based on cellulose fibers. This is a promising alternative for plastics replacing especially in the packaging industry and for improving the material efficiency by lowering the density and weight of products. In addition, the foam forming technology enables to use a large variety of recyclable or renewable materials and by replacing of water with foam gives the possibility to use a higher consistency after forming and wet pressing with important reduction of water and drying energy consumptions.

The mechanical strength, especially the compressive modulus, is a very important property for the cellulose foam formed materials (especially for those used as protective and cushioning packaging). Therefore, the main challenges to produce of these material grades using foam forming process is associated with identification of the technical solutions to compensate the strength loss caused by reduced density as well as to improve their water sensitivity.

It is very important to understand the interactions between foam and fibers, how fibers affect foam properties, and to find the optimal combination, distribution, and treatments of raw materials and additives to improve the mechanical strength of obtained products.

The literature survey shows that the strength of cellulose foam formed materials can be improved by cellulose fibers refining. In this case, the inter-fibers bonds and contact area between fibers is increased as result of external fibrillation and a more uniform foam material is obtained with improved structural integrity. Additionally, by including in the fibrous composition of the micro(nano-)fibrillated cellulose structures and polymeric additives (i.e., polyacrylamide, alcohols, polysaccharides etc.), it is possible to obtain the materials with strong mechanical strength in spite of its light weight and high porosity. However, to meet the environmental requirements (i.e., biodegradability) the utilization of biopolymers instead of synthetic ones could be another challenge in the field of foam formed cellulose materials.

Besides fibrous components, the charge and type of surfactant is of high interest in foam forming process. The electrical charge of a surfactant can interact with those of fibers and polymers, contributing to their bonding. Further, an appropriate compatibility of surfactant with other process additives must to be ensured. Some additives such as polyvinyl alcohol can be used both as surfactant and as strength additive to functionalize the formed structure. In this case the amount of other chemicals in process may be reduced.

In terms of environmental impact, the foam formed cellulose materials could be a sustainable alternative to the existing protective and cushioning packaging based on-plastic foams and expanded or extruded polystyrene and can considerably reduce the costs and issues associated with their recycling or disposal.

## Figures and Tables

**Figure 1 polymers-14-01963-f001:**
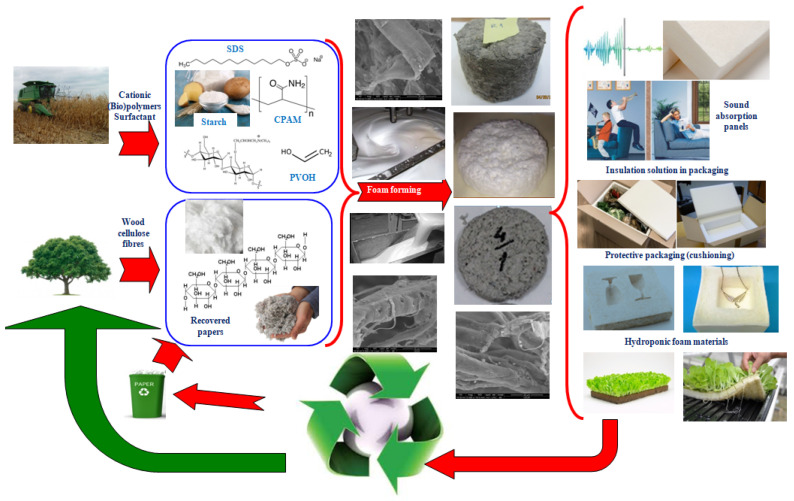
Overview on the area of application of foam formed cellulose materials.

**Figure 2 polymers-14-01963-f002:**
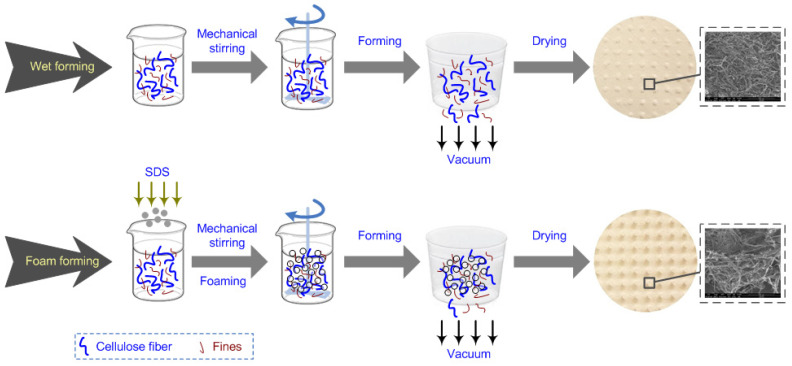
Comparison between foam forming and water forming processes.

**Figure 3 polymers-14-01963-f003:**
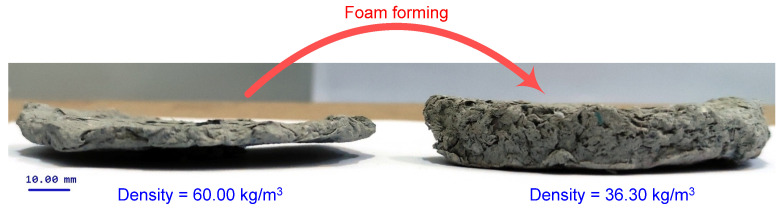
Cellulose materials from recycled fibers: water forming (**left**); foam forming (**right**).

**Figure 4 polymers-14-01963-f004:**
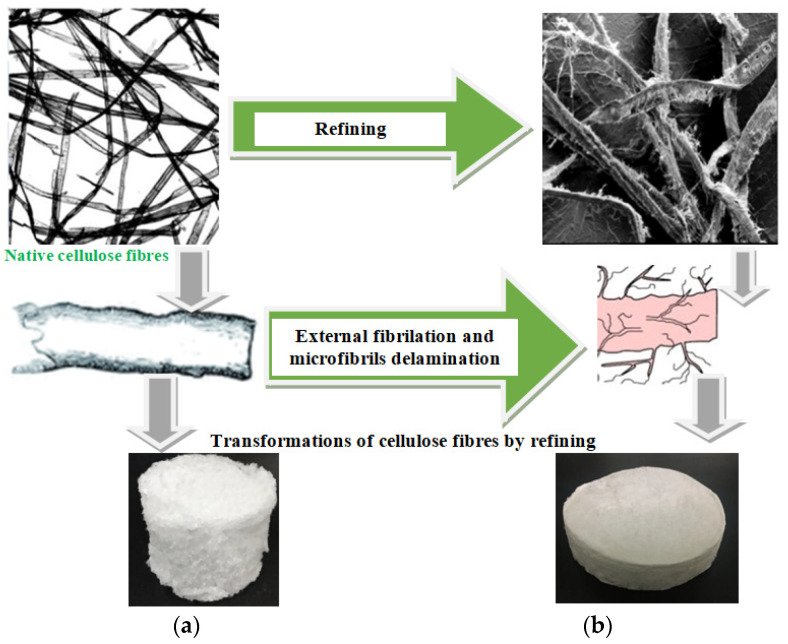
The effect of refining process on the structure and mechanical integrity of cellulose foam materials. (**a**) appearance of foam formed cellulose materials without refining; (**b**) appearance of foam formed cellulose materials after refining at 60°SR.

**Figure 5 polymers-14-01963-f005:**
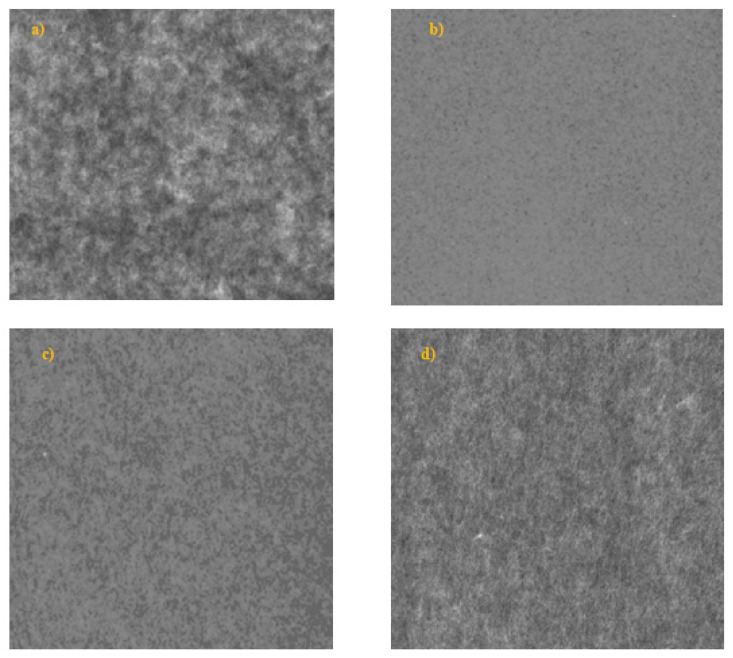
The influence of fiber length on the structure of foam formed cellulose materials. (**a**) water formed material from 100% hardwood pulp, fibre length = 1.0 mm; (**b**) foam formed material from 100% hardwood pulp, fibre length = 1.0 mm; (**c**) foam formed material from 100% softwood pulp, fibre length = 2.3 mm; (**d**) foam formed material from 80% softwood pulp + 20% Lyocell, fibre length = 3.0 mm.

**Figure 6 polymers-14-01963-f006:**
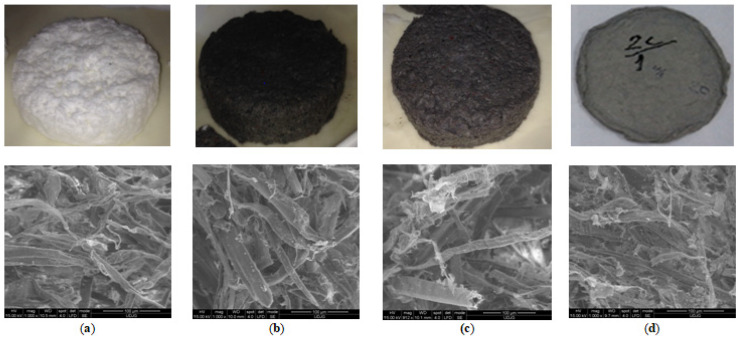
Internal structure of foam formed cellulose materials with high recycled fibers content. (**a**) Foam formed materials from 100% bleached cellulose fibers; (**b**) foam formed materials from 100% recycled cellulose fibers; (**c**) foam formed materials from 50% bleached cellulose fibers+50% recycled cellulose fibers; (**d**) water formed materials from 100% recycled fibers.

**Figure 7 polymers-14-01963-f007:**
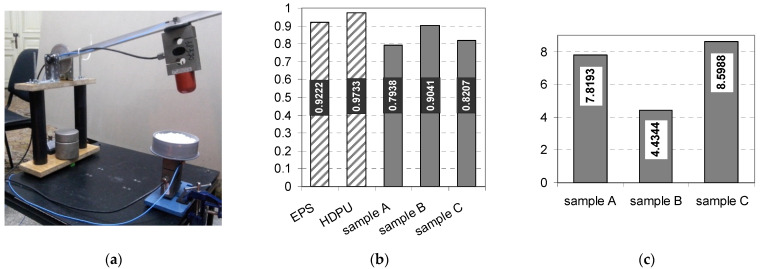
Shock absorption properties of foam formed cellulose materials. (**a**) Experimental setup used for impact tests; (**b**) restoring coefficient; (**c**) sample deformation, [mm]. EPS, expanded polystyrene foam; HDPU, high density polyurethane foam; Sample A, foam formed materials from 100% bleached cellulose fibers; Sample B, foam formed materials from 100% recycled cellulose fibers.

**Figure 8 polymers-14-01963-f008:**
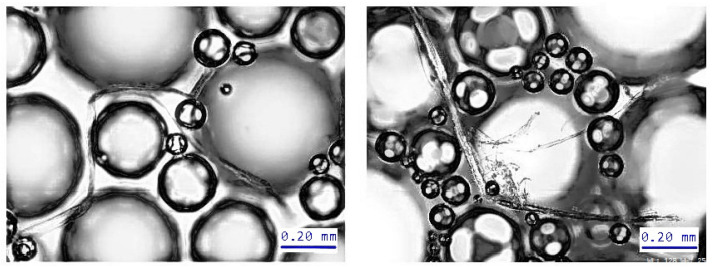
Cellulose fibers in foam. The fiber-foam bubbles contacts at high charge of surfactant. Re-print according to ref. [90].

**Table 1 polymers-14-01963-t001:** The packaging performance of cellulose materials *vs* petroleum-based alternatives.

	Molded Pulp	Foam Formed Cellulose Materials	Corrugated Board	Plastic Foams (eps)
Material	Wood/plant cellulose fibers/recycled fibers/waste	Wood/plant cellulose fibers/recycled fibers/waste	Wood/plant cellulose fibers/recycled fibers	Petroleum based
Sustainability	Recyclable/biodegradable/ compostable	Recyclable/biodegradable/ compostable	Recyclable/biodegradable/compostable	Non-biodegradable Non-recyclable Release toxic gas when burning [1,4]
Cushioning performance	Excellent vibration and cushioning properties Limited to package of products with low weight and volume [1,4]	Excellent vibration and cushioning properties	Inconsistent vibration and cushioning properties Not sufficiently soft [1,4]	Good vibration and cushioning properties
Shipping and storage	Easily nests [4]	Easily nests [4]	Require labor and assembly	Does not nest [4]
Climate tolerance	Is not affected by extreme external conditions	Not affected by extreme temperature Moisture proof by use of hydrophobic cellulose fibers [5,6]	Humidity affects performances	Temperature affects brittleness [4]
Durability and mechanical strength	Good	Can be improved by refining of cellulose fibers [7], using of strength additives [8] and surfactant type [9]	Good	High mechanical; strength and durability
Price	High	Low	Low	High

**Table 2 polymers-14-01963-t002:** Water *vs* foam formed cellulose materials.

	Water Formed Cellulose Materials	Foam Formed Cellulose Materials	Reference
Wet formation	High fibers flocculation	The foam prevents the flocculation of fibers. The fibers are locked between the foam bubbles and thus do not flock during network formation	Radvan et al. [26,27] Smith et al. [28,29]
	Low consistency of pulp suspension in head box	High consistency of pulp suspension in head box	Punton et al. [30,31] Poranen et al. [45]
	High water and drying energy consumptions	Economy of water and energy drying consumptions	
Structure of fibrous network	High density materials with 2D structure	Lightweight and low-density materials with 3D structure	Punton et al. [30]
The compression behavior	Denser materials with low recovery after compression	Bulking structure with better recovery after compression	Jarvinen et al. [46]
Fibrous raw materials	Cellulose fibers (bleached/unbleached pulp, secondary cellulose fibers)	Flexible long fibers, man-made fibers, recycled papers, lignocellulose agro-wastes	Nechita et al. [47] Beluns et al. [43]

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
