# Peer review of "Overview on Foam Forming Cellulose Materials for Cushioning Packaging Applications"

_polymers, 2022, doi:10.3390/polym14101963_

Round 1

Reviewer 1 Report

This work reviews the foam-forming cellulose materials for cushioning packaging applications. Compared to plastic foams as well as paper products (e.g. corrugated board, honeycomb board, moulded pulp), the significance of foam-forming cellulose materials (FFCM) has been primarily discussed in the first section. Then the basic introduction to the FFCM has been brought in. Most of the space is reserved for the words about technical parameters with influence the strength and structural properties of FFCM. The topic is, without doubt, important since plastic pollution has become a public issue in the world. However, the structure of the review paper needs to be optimized for improving the quality of this paper. In a word, a major revision is required.

Suggestions for the revision are as follows:

  1. The writing style of section 2 and section3 seems quite different. In my preference, section 2 will be more received since it exhibits the comments of authors in this field, and one can catch the key point easily. On the other hand, in section 3, it seems that we have read papers in this field in categories of four with the assistance of the detailed introduction of one paper within one paragraph. So one can hardly find the important information or work. Therefore, it is suggested that authors should “process” the research introduced in section 3, making them valuable information on the connection and distinction of papers in these fields.
  2. By classifying with technical parameters, it is also suggested to focus on the issues of improving the performance of FFCM. For example, a short discussion about solving the issue of moisture absorption and the resulting weakening of FFCM is welcomed. Similar issues can be of improving Young’s modulus, improving the compressive energy, decreasing the density, etc.
  3. The effect of polymer additives (e.g. polyacrylamide) on biodegradability needs to be discussed somewhere. Since FFCM can serve as an alternative to the non-biodegradable polymer (e.g. expanded PS).
  4. It is encouraged to use tables for summarizing and comparing papers, especially for numerical results.
  5. The origin of Figure 1, Figure 2, Figure 3, and Figure 6 needs to be referenced. The figure caption of Figure 1 and Figure 3 needs to be checked.
  6. There is something wrong with the serial number of “5. Conclusions”, section 4 is missing.
  7. The website mentioned in Ref 15 cannot be opened (error 404).
  8. The format of references needs to be carefully revised: e.g., the question mark in Ref 17 and 83; the journal name (including abbreviation) in Ref 74, 80, 84; the missing DOI information in Ref 75-77.
  9. The writing of English throughout the manuscript should be improved by a native English speaker. Low-level language errors should be avoided, such as in the ABSTRACT, “In this review are described the …”; in page 2 line 46: “…when is used as…”, and so many like these errors.

Author Response

Dear reviewer,

Thank you very much for your valuable review, comments, and opinions which helped us to improve the manuscript.

Taking into account all the comments and requirements of reefers, the manuscript was completely re-evaluated in order to respect the entirely ensemble of requirements requested by you and by the other reviewers. Practically, the work has been re-considered according to the reviewers' recommendations, resulting in the version that we have uploaded on platform, the differences from the original work being clearly marked with red colour.

However, you will find attached our answers for your questions and comments.

The authors

Reviewer 2 Report

Comments to authors

The review “Overview on Foam Forming Cellulose Materials for Cushioning 2 Packaging Applications” by Nechit and Nastac describes the techniques and properties of cellulosic materials for packaging applications. Although the topic of review is interesting but it has some limitations which are mentioned below in comments. After the major revision manuscript can be accepted for publication. There are few suggestions for authors to improve the manuscript for publication.

  1. Through out the whole manuscript there are too many short paragraphs that needs to me linked and merged in a logical way.
  2. Abstract needs a major revision. It should explain the benefits of packaging materials, methods to produce cellulose based foams and its properties and applications briefly.
  3. The overall language of the manuscript needs to be revised. In the current form it is hard to understand the meaning of various sentences like line 75, 119 and so on.
  4. If figure.1 is reproduces using reference 16 and 17 mentioned at line 81 then also metion references in figure legend.
  5. At some places the word “et al” is written as “collab” line 368, 604 and 572. Every time published manuscript with more authors is written as “et al” not as “collab”. Replace in whole manuscript.
  1. The Introduction and discussion can be strengthened by citing some recent relevant articles. You may read the following relevant articles.
  • Biobased materials for active food packaging: A review. 2022, Food Hydrocolloids 125:107419.
  • Development and characterization of Yeast-incorporated antimicrobial cellulose biofilms for edible food packaging applications. Polymers 13(14):2310
  1. It is recommended to check whole manuscript for grammatical, typos and spelling mistakes.
  2. For better presentation of the figures, unify the font size and style in all figures. Especially in figure 2 the blue text is not readable at first glance, change it to black and times new roman and increase the text size.
  3. Provide a representative Table that summarizes the effect of different parameters on foam properties and its applications or design a Table that shows which additive enhance the which special property of foam and used for what kind of packaging application along with reference.

Author Response

(The authors gave the same response as above.)

Reviewer 3 Report

The paper is sound. It is well discussed. The topic is perspective and very important for the field. There is a lack in review for cellulose formas and aerogels.

However, some 2021-2022 literature references in the field need to be used in the text. For example, 

DOI: 10.3390/polym12122779

10.1016/j.jaap.2021.105359
DOI: 10.1016/j.rser.2021.111483

DOI: 10.1016/j.indcrop.2021.113780

10.1016/j.indcrop.2021.114424

Many other sources from 2021 and 2022. Minor revision is recomended.

Author Response

(The authors gave the same response as above.)

Round 2

Reviewer 2 Report

Need to further work to organize and link the paragraphs 

Author Response

Dear reviewer,

Thank you very much for your valuable suggestion which helped us to improve the manuscript.

As response at your recommendation, the manuscript was carefully read and some paragraphs were linked, particularly  based on  the same issue discussed.   As result an appropriate structure of the manuscript was obtained.  The changes are green colour marked in the new revised manuscript.

The authors

Reviewer 3 Report

The paper was modified according to comments. It is acceptable for publishing.

Author Response

Dear reviewer,

Thank you for your valuable comments and suggestions.

The authors